# Glucose-Lowering Drugs and Primary Prevention of Chronic Kidney Disease in Type 2 Diabetes Patients: A Real-World Primary Care Study

**DOI:** 10.3390/ph17101299

**Published:** 2024-09-29

**Authors:** Antonio Rodríguez-Miguel, Beatriz Fernández-Fernández, Alberto Ortiz, Miguel Gil, Sara Rodríguez-Martín, Gema Ruiz-Hurtado, Encarnación Fernández-Antón, Luis M. Ruilope, Francisco J. de Abajo

**Affiliations:** 1Department of Biomedical Sciences (Pharmacology), School of Medicine and Health Sciences, University of Alcalá (IRYCIS), 28805 Alcalá de Henares, Spain; antonio.rodriguezmig@uah.es (A.R.-M.);; 2Department of Nephrology and Hypertension, IIS-Fundación Jiménez Díaz, Universidad Autónoma de Madrid, 28049 Madrid, Spain; bfernandez@fjd.es (B.F.-F.); aortiz@fjd.es (A.O.); 3RICORS2040, Institute of Health “Carlos III”, 28040 Madrid, Spain; 4Department of Medicine, Universidad Autónoma de Madrid, 28049 Madrid, Spain; 5BIFAP, Division of Pharmacoepidemiology and Pharmacovigilance, Spanish Agency for Medicines and Clinical Devices, 28022 Madrid, Spain; 6Cardiorenal Translational Laboratory, IIS-Imas12, University Hospital “12 de Octubre”, 28041 Madrid, Spain; gemaruiz@h12o.es (G.R.-H.);; 7Department of Physiology, School of Medicine, Universidad Autónoma de Madrid, 28049 Madrid, Spain; 8CIBER-CV, University Hospital “12 de Octubre”, 28041 Madrid, Spain; 9Clinical Pharmacology Unit, University Hospital “Príncipe de Asturias”, 28805 Madrid, Spain; 10School of Doctoral Studies and Research, European University of Madrid, 28040 Madrid, Spain

**Keywords:** chronic kidney disease, GLP-1 receptor agonists, glucose-lowering drugs, primary prevention, SGLT-2 inhibitors, type 2 diabetes

## Abstract

**Background/Objectives:** The burden of chronic kidney disease (CKD) is increasing, as is the prevalence of type 2 diabetes mellitus (T2DM). Post-hoc analyses of clinical trials support that sodium–glucose cotransporter-2 inhibitors (SGLT-2i) and glucagon-like peptide-1 receptors agonists (GLP-1RAs) prevent CKD in T2DM patients. **Methods:** We used the Spanish primary care database BIFAP to perform a retrospective cohort study with a nested case-control analysis to assess the incidence, risk factors, and the effect of glucose-lowering drugs (GLDs) on the primary prevention of CKD. **Results:** From a cohort of 515,701 T2DM subjects (2.75 million person-years), we found 89,075 incident CKD cases, yielding an overall incidence rate (95%CI) of 324.3 (322.1–326.5) per 10,000 person-years. In the nested case–control analysis, gout, hyperuricemia, and hyperkalemia were the factors showing the highest AORs. Long-term users (≥3 years) of GLP1-RAs and SGLT-2i, compared to other GLDs, showed a decreased risk for CKD (AOR = 0.85; 95%CI: 0.73–0.99 and AOR = 0.89; 95%CI: 0.74–1.08, respectively), and for incident CKD at KDIGO stages G3-G5 (AOR = 0.72; 95%CI: 0.56–0.94 and AOR = 0.64; 95%CI: 0.46–0.91, respectively). **Conclusions:** In a real-world primary care setting, the long-term use of GLP-1RAs and SGLT-2i, but not other GLDs, appeared to decrease the risk of incident CKD in T2DM, supporting a role in primary prevention of CKD.

## 1. Introduction

Around 850 million persons worldwide have chronic kidney disease (CKD) [1]. The global burden of CKD is increasing due to the aging of the population and the growing prevalence of type 2 diabetes mellitus, obesity, and hypertension [2]. The Global Burden of Disease study has estimated that CKD will become the fifth global cause of death by 2040 and the second by 2100 [3]. During the COVID-19 pandemic, CKD was the most associated risk factor with severe disease and mortality from COVID-19 [4]. CKD is diagnosed when the estimated glomerular filtration rate (eGFR) falls below 60 mL/min/1.73 m^2^ (chronic kidney insufficiency) or when albuminuria exceeds 30 mg/g of urinary creatinine, but also when there is other evidence of kidney injury persisting longer than 3 months with a negative impact on health [5]. The well-established thresholds for eGFR and albuminuria used by clinicians to guide the diagnosis of CKD are associated with an increased risk of CKD progression to failure requiring kidney replacement therapy, acute kidney injury (AKI), as well as all-cause and cardiovascular death [6]. Unfortunately, the diagnosis of CKD based on current diagnostic tools and criteria usually occurs very late, when over 50% of the renal mass has already been lost, which is equivalent to an eGFR of 60 mL/min/1.73 m^2^, and when there is no turning back in terms of renal function recovery [2,7]. Albuminuria may identify earlier stages of CKD, but this diagnostic test is underused so far, and in many countries, reimbursement is limited to patients with type 2 diabetes [8,9]. Despite recent advances in the treatment of CKD, neither renin-angiotensin system blockade nor sodium–glucose cotransporter-2 inhibitors (SGLT-2i), the current standard of therapy for diabetic and non-diabetic CKD, have been shown to restore the risk of kidney failure and premature death to baseline in any group of age and sex [10]. Moreover, kidney replacement therapy is still associated with a shortening of life expectancy of up to 40 years compared to subjects without and with extraordinarily high associated health costs [11].

Therefore, it is crucial to develop tools that allow the identification of subjects at risk of developing CKD and to design interventions for the prevention of CKD. In this regard, the successful use of primary prevention for cardiovascular disease has contributed to a global decrease in cardiovascular mortality in age-adjusted terms, which is in stark contrast to the increasing burden of CKD [3]. Recent clinical trials of novel glucose-lowering drugs (GLDs) showed the efficacy of these agents to slow the progression of CKD [10,12,13]. Also, post hoc analyses of cardiovascular clinical trials support the idea that these agents might also prevent new-onset CKD among patients with type 2 diabetes at high cardiovascular risk [10,14]. However, given the restrictive eligibility criteria of these trials, the underrepresentation of women, and the nature of participating centers (usually endocrinology units within specialized centers with clinical trial units) [12,15,16], it still remains to be seen whether these findings can be generalized to primary care settings, which would be the natural environment for primary prevention efforts, including CKD.

Large automated databases, such as BIFAP (“*Base de datos para la Investigación Farmacoepidemiológica en el Ámbito Público*”) [17], which comprises electronic health records of patients attended in primary care centers around Spain (real-world data) combined with the use of state-of-the-art methods in epidemiology and statistical learning, are key tools to help ensure whether the findings from clinical trials are replicated in real-life, as well as to help develop new predictive or prognostic models of CKD [7].

In the present study, we examine the epidemiology of CKD in a cohort of patients with type 2 diabetes extracted from BIFAP to better understand the drivers of incident CKD and to evaluate the impact of different GLDs in the primary prevention of CKD in real-world conditions.

## 2. Results

The general cohort consisted of 5,982,868 subjects, and from it, we identified a total of 515 701 patients who fulfilled the inclusion criteria and made up the T2DM cohort (Figure 1). The median age (IQR) of the T2DM cohort at the start of the follow-up was 62 (53–71) years, and 46.2% (238,096) were women.

### 2.1. Incidence of CKD in the T2DM Cohort

Over the study period, 89,075 new cases of CKD were identified in the T2DM cohort. The median (IQR) time from entry to new onset of CKD was 3.42 (1.57–6.10) years, 44,016 (49.4%) were women, and the median (IQR) age at CKD detection was 72 (64–79) years. 

The total follow-up time of the T2DM cohort was nearly 2.75 million person-years, resulting in an overall CKD incidence rate (95%CI) of 324.3 (322.1–326.5) per 10,000 person-years for the period 2005–2019. The incidence rate of CKD increased with age and was higher in women than in men from the age of 70 onwards (Figure 2).

Crude incidence rates per 10,000 person-years overall and by different comorbidities are shown in Table 1. All comorbidities evaluated were associated with an increased risk of developing CKD, with heart failure and peripheral artery disease being the factors with the highest incidence rates. 

Incidence rates of CKD by calendar year remained fairly constant throughout the study period overall and in all 10-year age groups (Figure 3).

### 2.2. Nested Case–Control Analysis

The socio-demographic and clinical characteristics of cases and controls are detailed in Table 2. A total of 89,075 incident CKD cases and 442,216 controls were analyzed (Figure 1). Among cases, 40.8% were detected at stage G1–G2, 42.4% at G3a, 6.47% at G3b, 0.87% at G4, and 0.20% at G5. Overall, 44,447 (49.9%) were first detected with CKI (stage ≥ G3). In some cases (9.28%), there was no information on eGFR to assign the stage at the index date.

One or more isolated records of abnormal eGFR (16,160; 18.1%) and/or positive proteinuria/albuminuria (13,996; 15.7%), never confirmed in a second consecutive measurement, were detected more often in cases than in controls, yielding an OR (95%CI) of incident CKD of 1.82 (1.78–1.85) for isolated abnormal eGFR, and 1.45 (1.42–1.48) for isolated positive albuminuria/proteinuria (Table 2). Other risk factors associated with incident CKD included history of gout (OR = 2.38; 95%CI: 2.32–2.45), hyperuricemia (OR = 2.30; 95%CI: 2.26–2.34), hyperkalemia (OR = 2.18; 95%CI: 2.08–2.27), hypertension (OR = 1.65; 95%CI: 1.62–1.68), heart failure (OR = 1.64; 95%CI: 1.61–1.68), hyperparathyroidism (OR = 1.61; 95%CI: 1.50–1.72), PAD (OR = 1.42; 95%CI: 1.38–1.45), current smoking (OR = 1.40; 95%CI: 1.38–1.43), dyslipidemia (OR = 1.36; 95%CI: 1.33–1.38), ischemic heart disease (including acute myocardial infarction and angina; OR = 1.35; 95%CI: 1.32–1.38), atrial fibrillation (OR = 1.34; 95%CI: 1.31–1.36), obesity (OR = 1.32; 95%CI: 1.29–1.36), hematuria (OR = 1.28; 95%CI: 1.25–1.31), and stroke (including ischemic, hemorrhagic and transient ischemic attack; OR = 1.15; 95%CI: 1.12–1.17) (Table 2).

The results of the association of the current use of different GLDs with incident CKD and the impact of duration of use are shown in Figure 4 and Figure 5, and in more detail in Appendix A. A moderately increased AOR was observed with the long-term use (≥3 years) of most GLDs. In contrast, a significantly lower AOR was observed with the long-term use of GLP-1RAs (AOR_≥3 years_ = 0.85; 95%CI: 0.73–0.99) and a non-significant downward trend with SGLT-2i (AOR_≥3 years_ = 0.89; 95%CI: 0.74–1.08) (Figure 4). Of note, the negative associations found with the long-term use of GLP-1RAs and SGLT-2i were significantly more marked when the outcome variable was restricted to incident CKI (stages G3–G5): AOR (95%CI) was 0.72 (0.56–0.94) and 0.64 (0.46–0.91), for GLP-1RAs and SGLT-2i, respectively (Figure 5). Such negative associations with SGLT-2i and GLP-1RAs were obtained despite presenting the longest median follow-up time (since the initiation of any GLD until the index date), indicating that the time of progression of type 2 diabetes was longer in them than with other GLDs (Figure 4 and Figure 5). Likewise, the negative association observed with the short-term use of alpha-glucosidase inhibitors, not sustained in the long-term, can be interpreted as patients on these drugs having early stages of the disease.

## 3. Discussion 

The results from the present study showed that, in a real-world primary care setting, the long-term use of GLP-1RAs and SGLT-2i in type 2 diabetes patients was associated with a significantly decreased risk of new-onset CKI (CKD at stages G3–G5). A similar pattern was observed when all stages of CKD were considered, but SGLT-2i did not reach statistical significance. These results support the hypothesis that these drugs have the potential for primary prevention of CKD in patients with type 2 diabetes. These results confirm in a real-world setting the findings obtained in post hoc analyses of secondary endpoints from randomized clinical trials [12,14,15,16,18]. Regarding the epidemiology of CKD in primary care and among patients with type 2 diabetes, we found the following: (1) over the study period, the incidence rate of CKD was stable overall; (2) in patients older than 70 years, the incidence rate was higher in females than in males; and (3) the factors more strongly associated with incident CKD were the antecedents of gout, hyperuricemia, hyperkalemia, hypertension, heart failure, hyperparathyroidism, and prior isolated abnormal values of eGFR or proteinuria/albuminuria.

Having a prior record of isolated low eGFR or high albuminuria values was an important risk factor for incident CKD. This may reflect that these patients were on the brink of fulfilling CKD criteria and may have had transient low eGFR or higher albuminuria episodes triggered by superimposed acute illness or drugs (e.g., nephrotoxic agents) or modified by the addition of antiproteinuric or antihypertensive agents. Episodes of AKI are known to accelerate CKD progression [19]. We found that hypertension, heart failure, gout or hyperuricemia, hyperkalemia, or hyperparathyroidism were associated with a higher risk of developing CKD. Hypertension and heart failure are known risk factors for kidney disease, while hyperuricemia, hyperkalemia, or hyperparathyroidism may be either risk factors or indicators of already compromised kidney function [20]. Moreover, hyperkalemia may preclude the optimization of renin-angiotensin system blockade, a key kidney protective maneuver [21]. It is interesting to note that both SGLT-2i and GLP-1RAs may have a beneficial effect on some of these risk factors, and SGLT-2i may also decrease serum uric acid levels and the risk of hyperkalemia [22,23,24].

Our study provides a glimpse of GLDs usually managed in primary care in Spain. During the study period, patients with type 2 diabetes were overwhelmingly treated with drugs that have not been demonstrated to improve cardiovascular and kidney outcomes in randomized controlled trials. Of note, users of SGLT-2i and GLP-1 RAs had the longest time of duration of type 2 diabetes, on average, so they were likely at a higher risk of CKD. If SGLT-2i and GLP-1RAs did not have any effect, we should have observed an increased risk of CKD as we did among the users of some of the remaining GLDs, especially in the long-term; the latter may have reflected the lack of such improvement of kidney outcomes. GLP-1 RAs and SGLT-2i were among the least commonly prescribed GLDs, likely due to their more recent marketing authorization. However, in clinical trials, SGLT-2i improved cardiovascular and kidney outcomes in patients with or without type 2 diabetes [10,25,26], and similar results were observed with the use of GLP-1 RAs in the setting of type 2 diabetes or obesity [14,23,27,28,29,30]. Additionally, recent Guidelines from KDIGO and the American Diabetes Association recommend SGLT-2i as first-line therapy for patients with diabetes and CKD, while GLP-1RAs are second-line drugs [26]. Even more interesting were the findings from post hoc analyses of cardiovascular safety trials in patients with type 2 diabetes at high cardiovascular risk, where canagliflozin, dapagliflozin, and empagliflozin decreased incident CKD among patients with eGFR and albuminuria within the normal range at baseline [12,15,16]. Moreover, kidney protection was observed in post hoc analyses of clinical trials of GLP-1RAs in which many patients had renal impairment at baseline [31]. Recently, a dedicated trial of kidney outcomes with semaglutide (FLOW trial) was stopped early by the sponsor due to efficacy, adding strong evidence on renoprotection of GLP1-Ras [13,32]. However, it is important to emphasize that all patients had various degrees of kidney damage, with 79.6% presenting an eGFR lower than 60 mL/min/1.73m^2^; thus, the primary prevention of CKD was not really addressed [13]. In this context, the results of our study support the renoprotective efficacy of SGLT2i and GLP-1-RAs and add the important novelty of being focused on primary prevention of CKD, as all patients had no evidence of kidney damage at the time of inclusion. Further, our study provides evidence of renoprotection of SGLT-2i and GLP-1RAs in a real-world primary care setting in an unselected population that encompassed a higher proportion of women and a longer follow-up than in clinical trials. Nevertheless, some limitations should be acknowledged, starting with the observational and retrospective nature of the study, leaving room for residual confounding. Additionally, neither albuminuria nor proteinuria were exhaustively recorded, which may have led to a certain under-recording of CKD in its early stages. Also, around 10% of cases were detected by albuminuria/proteinuria or a recorded diagnosis of CKD but with no data on eGFR or creatinine at the index date, and we were unable to assign them a KDIGO stage. Although we were unable to confirm that the two consecutive abnormal levels were separated by at least 90 days, in previous analyses of our cohort, we found that levels of serum creatinine or eGFR were recorded in the database for each subject every year, on average, so we could be confident that such time interval between measurements also occurred in our cases. However, this represents current real-world clinical practice, and our study allows for identifying weaknesses of primary care practice for the early detection of CKD, pointing to areas of improvement. Of note, given that the cohort entry criteria were based on a recorded diagnosis of CKD in the electronic health records or meeting eGFR or proteinuria/albuminuria criteria of CKD, our definition of CKD in the present study was indeed more restrictive than the KDIGO definition that also includes imaging (e.g., polycystic kidney disease), histological, or other evidence of kidney injury as diagnostic criteria for CKD. However, the weaknesses are offset by the strengths: a real-world and primary care setting in several Autonomous Communities from Spain, in an extensive population of nearly half a million subjects with type 2 diabetes, and with a large follow-up of more than 2.7 million person-years. In this regard, we included subjects on treatment with SGLT-2i and GLP-1RAs with durations of use that exceed the duration of most clinical trials (≥3 years) and suggest that the largest benefit is obtained in the long-term. Additionally, to confirm CKD, we required at least two consecutive pathological values of either eGFR or albuminuria/proteinuria, which differs from other studies that used just a single abnormal eGFR or albuminuria values.

## 4. Patients and Methods

### 4.1. Source of Information

We used the Spanish electronic database BIFAP, which comprises pseudonymized electronic health records of patients attended by primary care physicians and pediatricians from the National Health System in 9 Autonomous Communities (Aragón, Asturias, Canary Islands, Cantabria, Castilla y León, Madrid, Murcia, Navarra, and Castilla-La Mancha). The information available in BIFAP is recorded by the physicians in their daily routine and includes demographic data, diagnoses, specialist referrals, clinical notes in free text, drug prescriptions, and other relevant health data (lifestyle habits or laboratory results, among others) [17]. This information is updated every six months. BIFAP is funded and maintained by the Spanish Agency for Medicines and Medical Devices (AEMPS) and has been validated against other European databases and through many pharmacoepidemiologic studies in different areas, including cardiovascular [17,33]. In this study, we used the version that included health data up to 31 December 2019.

### 4.2. Design and Study Population

A retrospective cohort study with a nested case–control analysis was carried out. From 2005 to 2019, all patients of any sex and age were included when they fulfilled a minimum registry of 1-year with their primary care physician and had at least two valid records of serum creatinine and/or eGFR during the follow-up, being the first in the normal range (eGFR: ≥60 mL/min/1.73 m^2^; serum creatinine: 0.1–1.1 mg/dl in women and 0.1–1.3 mg/dl in men). The date of the first normal record was considered the entry date in the general cohort. Patients were excluded if, at a time prior to the entry in the cohort, they had a record of cancer (except non-melanoma skin) in the previous 3 years; a record of eGFR, albuminuria, or proteinuria outside the normal range; or a record of CKD, dialysis, or kidney transplantation. Patients with a previous record of AKI, unspecified kidney disease, or other kidney diseases were not excluded from the cohort but were identified for further analysis.

Within the general cohort, we identified those patients with type 2 diabetes, aged 18–90, of any sex, and on treatment with GLDs to make up the so-called type 2 diabetes mellitus cohort (T2DM cohort). For that purpose, we looked for patients who had at least one prescription of a GLD throughout the study period, with the date of the first prescription as the entry date in the T2DM cohort. Patients with a recorded diagnosis of type 2 diabetes without prescriptions of GLDs were not included, as most probably they were in an early phase of the disease and had a presumably lower risk profile than treated patients. Also, patients with insulin in monotherapy were excluded as they were likely to present with type 1 diabetes. 

### 4.3. Case Definition

Cases were identified as those with a record of eGFR < 60 mL/min/1.73 m^2^ based on the 2009 creatinine-based CKD-EPI formula in a race-agnostic manner [34] and/or positive albuminuria (>30 mg/24 h, >30 mg/g urinary creatinine) or positive proteinuria on urine test strips, which was confirmed in, at least, a second consecutive record of either out-of-range eGFR or positive proteinuria/albuminuria at a later date. If confirmed, the patient was considered a case on the date of the first abnormal record, that is, the index date of the case. By contrast, if a pathological record was followed by a normal one, the patient was not considered a case. The stage of CKD was classified according to the following KDIGO categories [6]: G1-G2, G3a, G3b, G4, and G5. Cases whose first abnormal record was albuminuria/proteinuria or a diagnosis of CKD and without a record of eGFR could not be classified. Some analyses were restricted to cases of chronic kidney insufficiency (CKI), defined as those with CKD at stages from G3a to G5. 

### 4.4. Follow-Up

All subjects were followed from the entry date into the T2DM cohort up to the occurrence of a confirmed case of CKD or any of the following censorship criteria: death from all causes or loss to follow-up for any other reason, turn 90 years old, or end of the study period (31 December 2019). The date of occurrence of any of the abovementioned events was considered as the end of follow-up. 

### 4.5. Nested Case–Control Study

For each CKD case identified in the T2DM cohort, we matched up to 5 controls for index date, sex, and age at index date (exact) following a risk-set sampling with replacement. This method of selecting controls is an incidence-density-based sampling and ensures that the more person-time each subject contributes to the cohort, the greater the probability of being selected as a control. It is also implicit that a subject could be sampled as a control for more than one case, though at different timepoints. This way, the measure of association obtained in the case–control study is an unbiased estimate of the hazard ratio that would be obtained in the underlying cohort [35], even in the presence of competing risks [36].

Exposure to GLDs was assessed in the year before the index date, and accordingly, we built independent variables for each GLD subgroup with three categories: (1) non-users, when no prescriptions of the GLD of interest were ever found; (2) current users, when the prescription of the GLD of interest lasted up to index date or ended within 1-year before index date; and (3) past users, when a prescription of the GLD of interest was found but ended more than 1-year before index date. The reference category for all comparisons was the non-use. As all subjects were users of GLDs, the non-use category can be interpreted as “current use of any other GLDs”. For clarity’s sake, results associated with the past use category were omitted. The effect of treatment duration among current users was also assessed by using the cumulative, continuous duration, which was the sum of days of all consecutive prescriptions of the same GLD subgroup, provided that no more than 60 days elapsed between the end of one prescription and the beginning of the next. The cumulative, continuous duration was evaluated in periods of less than 3 years and equal to or greater than 3 years (long-term use).

Potential confounding factors associated with the exposure to GLDs and incident CKD were selected by expert criteria, and their presence, as well as the use of comedications, was evaluated from the index date backward (Additional Methods). 

### 4.6. Statistical Analysis

Incidence rates with their 95% confidence intervals (95%CI) were estimated in the T2DM cohort using incident cases of CKD as the numerator, divided by person-time of follow-up. Incidence rates were also stratified by 10-year age bands, sex, calendar year, and by the following comorbidities: obesity, dyslipidemia, hypertension, acute myocardial infarction, angina, stroke, transient ischemic attack, peripheral arterial disease, and heart failure. Age, calendar year, and all comorbidities were treated as time-varying variables.

Trend analysis of the CKD incidence rates by calendar year was performed using linear regression. The existence of significant trend reversal points in the incidence rates throughout the study period was analyzed using piecewise linear regression by comparing the slopes of the two regression lines at both sides of each joint point. We detected a loss of information on cases leading to a shrink in the incidence rates for 2017, especially in older ages, so we used linear interpolation to estimate the missing values.

In the nested case–control design, we first performed a descriptive analysis of the socio-demographic and clinical characteristics as well as renal parameters (eGFR and proteinuria/albuminuria) of cases and controls. Qualitative variables were expressed as absolute and relative frequencies (percentages), and quantitative variables as median and interquartile range (IQR). The association between the variables of interest and incident CKD (overall and among subjects with CKI) was assessed through a conditional logistic regression model fitted to estimate the odds ratios (OR) and their 95%CIs, adjusted by age and sex by design, and the fully adjusted OR (AOR) including all potential confounding factors (Additional Methods).

All analyses were performed using Python 3.11 (Python Software Foundation 2022) and STATA 17/MP (StataCorp. College Station, TX, USA). A *p*-value < 0.05 was set as statistically significant.

## 5. Conclusions

In a primary care real-world setting, the prolonged use of SGLT-2i and GLP-1RAs in patients with type 2 diabetes was associated with a decreased risk of incident CKD, in particular among KDIGO stage ≥ G3, a finding that is in line with post hoc analyses of clinical trials of these agents. In addition, the majority of the remaining GLDs analyzed did not show evidence supporting their use for primary prevention or to control the progression of CKD, especially in the long-term. Overall, these findings support the feasibility of primary prevention of CKD in the primary care setting, as illustrated by a specific high-risk population like the one studied. Current guidelines address the primary prevention of cardiovascular disease in patients with type 2 diabetes, as well as the primary prevention of type 2 diabetes, but not the primary prevention of CKD [37]. In fact, they focus on patients with type 2 diabetes who already have CKD and recommend the initiation of GLDs with proven kidney protective effects after CKD has been diagnosed [26], while our findings support the generalization of the prescription of GLDs with kidney protective action for the preservation of kidney health to a wider population with type 2 diabetes.

## Figures and Tables

**Figure 1 pharmaceuticals-17-01299-f001:**
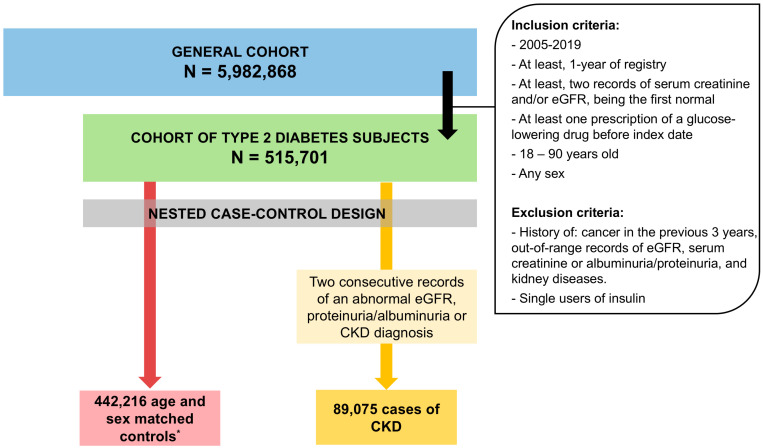
Flowchart of patient selection. eGFR: estimated glomerular filtration rate; CKD: chronic kidney disease. * Controls were selected using an incidence-density sampling with replacement, which allows a subject to act as the control for more than one case. This explains that the sum of cases and controls exceeds the number of patients in the type 2 diabetes cohort (see Section 4.5, “Nested case-control study” in Section 4, “Methods”).

**Figure 2 pharmaceuticals-17-01299-f002:**
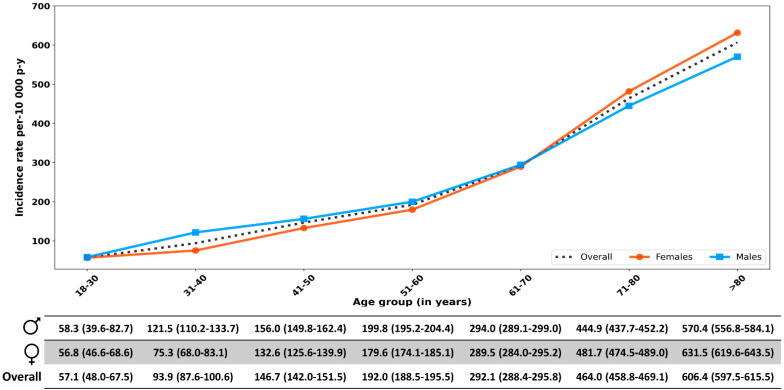
Incidence rates of chronic kidney disease by age and sex. p-y: person-years.

**Figure 3 pharmaceuticals-17-01299-f003:**
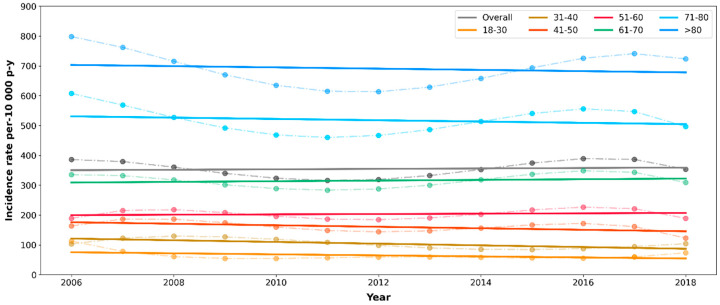
Trends in the incidence rates of chronic kidney disease by 10-year age bands. β_overall_ (95%CI) = −4.22 (−13.5, 5.07); β_18–30_ (95%CI) = −0.47 (−4.75, 3.80); β_31–40_ (95%CI) = −2.84 (−4.91, −0.77); β_41–50_ (95%CI) = −3.65 (−6.96, −0.34); β_51–60_ (95%CI) = −2.28 (−5.56, 0.99); β_61–70_ (95%CI) = −1.72 (−8.68, 5.24); β_71–80_ (95%CI) = −8.56 (−20.1, 2.96); β_>80_ (95%CI) = −10.0 (−26.5, 6.52).

**Figure 4 pharmaceuticals-17-01299-f004:**
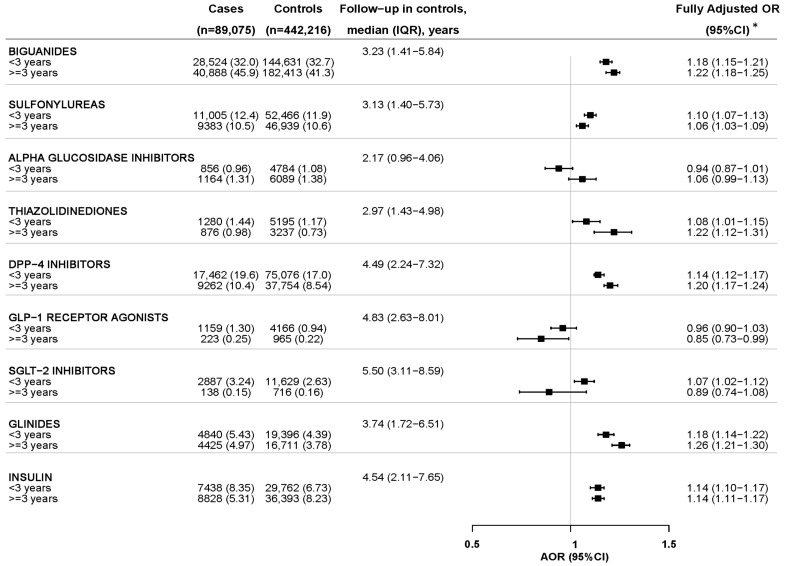
Incident chronic kidney disease overall and current use of glucose-lowering drugs by treatment duration. The reference comparison for each GLD is the non-use. Since all subjects are users of GLDs, the non-use category includes those users of any other GLD. IQR: interquartile range; CI: confidence interval; AOR: adjusted odds ratio. * Adjusted for all the variables in the Table 2, excepting CKD stage, plus the following comedication in the year before index date: antihypertensives (including alpha-adrenoreceptor antagonists, beta blocking agents, calcium channel blockers, angiotensin converting enzyme inhibitors, angiotensin II receptor blockers, and renin-inhibitors), diuretics (including high-ceiling, low-ceiling thiazides, low-ceiling excluding thiazides, direct potassium-sparing agents and, mineralocorticoid receptor antagonists), antiplatelet drugs (including COX-1 inhibitors, P2Y12 receptor blockers, others), oral anticoagulants (including vitamin K antagonists, direct thrombin inhibitors, direct factor Xa inhibitors), heparins, class I and III antiarrhythmics, nonsteroidal anti-inflammatory drugs, paracetamol, metamizole, symptomatic slow-action drugs for osteoarthritis, opioids, glucocorticoids for systemic use, proton-pump inhibitors, H2-receptor antagonists, immunosuppressants, benzodiazepines, antidepressants, antiepileptics, anti-Parkinson drugs, antipsychotics, vitamin D and calcium (alone or in combination), and colchicine.

**Figure 5 pharmaceuticals-17-01299-f005:**
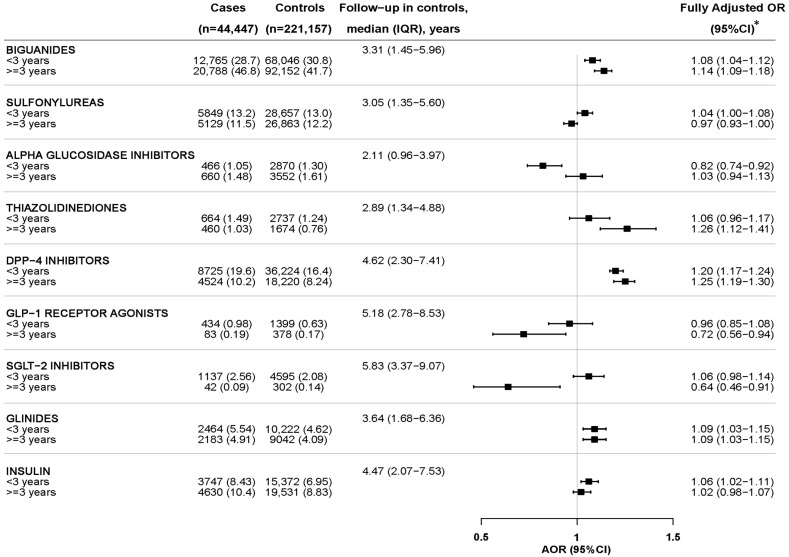
Incident chronic kidney insufficiency (G3–G5) and current use of glucose-lowering drugs by treatment duration. The reference comparison for each GLD is the non-use. Since all subjects are users of GLDs, the non-use category includes those users of any other GLD. IQR: interquartile range; CI: confidence interval; AOR: adjusted odds ratio. * Adjusted for all the variables in Table 2, excepting CKD stage, plus the following comedication in the year before index date: antihypertensives (including alpha-adrenoreceptor antagonists, beta blocking agents, calcium channel blockers, angiotensin converting enzyme inhibitors, angiotensin II receptor blockers, and renin-inhibitors), diuretics (including high-ceiling, low-ceiling thiazides, low-ceiling excluding thiazides, direct potassium-sparing agents and, mineralocorticoid receptor antagonists), antiplatelet drugs (including COX-1 inhibitors, P2Y12 receptor blockers, others), oral anticoagulants (including vitamin K antagonists, direct thrombin inhibitors, direct factor Xa inhibitors), heparins, class I and III antiarrhythmics, nonsteroidal anti-inflammatory drugs, paracetamol, metamizole, symptomatic slow-action drugs for osteoarthritis, opioids, glucocorticoids for systemic use, proton-pump inhibitors, H2-receptor antagonists, immunosuppressants, benzodiazepines, antidepressants, antiepileptics, anti-Parkinson drugs, antipsychotics, vitamin D and calcium (alone or in combination), and colchicine.

**Table 1 pharmaceuticals-17-01299-t001:** Incidence rates of chronic kidney disease overall and by different comorbidities in the type 2 diabetes mellitus cohort.

	Cases (%)	Person-Years	Incidence Rate (per 10 000 p-y) (95%CI)
**Overall**	89,075 (100)	2,746,449.0	324.3 (322.2–326.5)
**Overall, >40 years**	88,124 (98.9)	2,635,660.2	334.4 (332.2–336.6)
**Sex:** Males Females	45,059 (50.6) 44,016 (49.4)	1,466,109 1,280,340	307.3 (304.5–310.2) 343.8 (340.6–347.0)
**Obesity:** No Yes	36,079 (40.5) 52,996 (59.5)	1,220,921 1,525,528	295.5 (292.5–298.6) 347.4 (344.4–350.4)
**Dyslipidemia:** No Yes	38,238 (42.9) 50,837 (57.1)	1,278,445 1,468,004	299.1 (296.1–302.1) 346.3 (343.3–349.3)
**Hypertension:** No Yes	21,489 (24.1) 67,586 (75.9)	1,069,286 1,677,163	201.0 (198.3–203.7) 403.0 (400.0–406.0)
**AMI:** No Yes	83,145 (93.3) 5930 (6.66)	2,616,625 1,298,241	317.8 (315.6–319.9) 456.8 (445.2–468.6)
**Heart failure:** No Yes	82,320 (92.4) 6755 (7.58)	2,656,553 89,895.8	309.9 (307.8–312.0) 751.4 (733.6–769.6)
**Stroke:** No Yes	83,265 (93.5) 5450 (6.52)	2,630,715 115,734.2	317.9 (315.7–320.0) 470.9 (458.5–483.6)
**TIA:** No Yes	86,587 (97.2) 2488 (2.79)	2,692,669 53,779.9	321.6 (319.4–323.7) 462.6 (444.6–481.2)
**PAD:** No Yes	83,420 (93.7) 5655 (6.35)	2,638,262 108,187.5	316.2 (314.1–318.3) 522.7 (509.2–536.5)

p-y: person-years; CI: confidence interval; AMI: acute myocardial infarction; TIA: transient ischemic attack; PAD: peripheral artery disease.

**Table 2 pharmaceuticals-17-01299-t002:** Characteristics of cases and controls at index date.

	Cases (n = 89,075)	Controls (n = 442,216)	Age and Sex-Adjusted OR (95%CI) ^a^
**Female**, *n* (%)	44,016 (49.4)	218,238 (49.4)	Matched
**Age at index date**, in years, median (IQR)	72 (64–79)	72 (64–79)	Matched
**Follow-up to index date,** years, median (IQR)	3.42 (1.57–6.10)	3.26 (1.47–5.87)	1.02 (1.02–1.02)
**Alcohol abuse,** *n* (%)	19,530 (21.9)	98,870 (22.4)	0.97 (0.96–0.99)
**BMI,** *n* (%): <30 kg/m^2^/not recorded ≥30 kg/m^2^ (obesity)	31,993 (35.9) 57,082 (64.1)	186,814 (42.2) 255,402 (57.8)	Reference 1.32 (1.29–1.33)
**Smoking,** *n* (%): Non-smokers Ex-smokers Current smokers Not recorded	26,523 (29.8) 4359 (4.89) 25,784 (29.0) 32,409 (36.4)	147,870 (33.4) 29,406 (6.65) 104,678 (23.7) 160,262 (36.2)	Reference 0.85 (0.82–0.88) 1.40 (1.38–1.43) 1.12 (1.10–1.14)
**Isolated records of eGFR < 60 mL/min/1.73 m^2^ prior to index date,** *n* (%) >1 record	16,160 (18.1) 3178 (3.57)	49,280 (11.1) 7050 (1.59)	1.82 (1.78–1.85)
**Isolated records of positive albuminuria/proteinuria prior to index date,** *n* (%) >1 record	13,996 (15.7) 3564 (4.0)	50,763 (11.5) 7360 (1.66)	1.45 (1.42–1.48)
**CKD stage,** *n* (%): G1–G2 G3a G3b G4 G5 Not available ^b^	36,358 (40.8) 37,732 (42.4) 5763 (6.47) 777 (0.87) 175 (0.20) 8270 (9.28)	-	-
**Comorbidities at index date,** *n* (%): ^c^ Hypertension Dyslipidemia Atrial fibrillation Ischemic heart diseases Heart failure Gout (record or ULD use) Hyperuricemia (non-gout) Stroke PAD Hyperparathyroidism Osteoporosis Hematuria ^d^ Hyperkalemia ^d^	73,151 (82.1) 70,861 (79.6) 16,718 (18.8) 14,917 (16.8) 14,379 (16.1) 8263 (9.28) 41,010 (46.0) 13,226 (14.9) 9309 (10.5) 1050 (1.18) 11,950 (13.4) 8661 (9.72) 2988 (3.35)	328,205 (74.2) 328,464 (74.3) 66,095 (15.0) 57,795 (13.1) 47,708 (10.8) 26,681 (6.03) 131,817 (29.8) 58,634 (13.3) 33,989 (7.69) 3265 (0.74) 61,233 (13.9) 34,470 (7.79) 6960 (1.57)	1.65 (1.62–1.68) 1.36 (1.33–1.38) 1.34 (1.31–1.36) 1.35 (1.32–1.38) 1.64 (1.61–1.68) 2.38 (2.32–2.45) 2.30 (2.26–2.34) 1.15 (1.12–1.17) 1.42 (1.38–1.45) 1.61 (1.50–1.72) 0.96 (0.94–0.98) 1.28 (1.25–1.31) 2.18 (2.08–2.27)

OR: odds ratio; CI: confidence interval; IQR: interquartile range; BMI: body mass index; eGFR: estimated glomerular filtration rate; CKD: chronic kidney disease; ULD: urate-lowering drugs; PAD: peripheral artery disease. ^a^ Adjusted for age and sex by design; ^b^ cases detected by albuminuria/proteinuria or a recorded diagnosis of CKD but with no data on eGFR or creatinine at index date; ^c^ the category of reference is the absence of the comorbidity; ^d^ recorded by the physician as such.

## Data Availability

Data from BIFAP is restricted to non-profit organizations and independent researchers, so authors are not allowed to openly make them available. However, data could be available by reasonable request from any organization or researcher fulfilling those restrictions to the corresponding author, provided that the owner of BIFAP (the AEMPS) specifically authorizes the data transfer.

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
