# Peer review of "Glucose-Lowering Drugs and Primary Prevention of Chronic Kidney Disease in Type 2 Diabetes Patients: A Real-World Primary Care Study"

_pharmaceuticals, 2024, doi:10.3390/ph17101299_

Round 1

Reviewer 1 Report

Comments and Suggestions for Authors

This is a real-world study that illustrates the beneficial renal effects of SGLT2i and GLP1ra on primary prevention of CKD in patients with T2D. The evidence looks strong because of a large number of participants. However, I have some concerns and suggestions related to the study.

1. To design of the study (Fig. 1):

1.1. The authors did not noted the period between two consequent records of abnormal eGFR and UACR. Please ensure that it was at least 3 months according to the CKD definition by KDIGO.

1.2. The manuscript does not contain such important outcomes as death etc.

2. To statistical procedures: The authors analyzed a lot of data. May be it could be reasonable to count p-value for AOR and analyze the significance assuming multiple testing problem.

4. To interpretation of results: The authors proved the beneficial effects of SGLT2i and GLP1ra. However, the manuscript contains some questionable results, such as harmful effects of all other GLDs on CKD incidence. Why did the authors include these GLDs to analysis while they aimed to study effects of SGLT2i and GLP1ra? What could be the possible cause of this observation? Which flaws in the design of the study or statistical analysis could bias the results?

5. To presentation (Tab. 2): It is not clear what means the lines 1 (1-1) and 1 (1-2). Please, explain.

Author Response

Reviewer 1

This is a real-world study that illustrates the beneficial renal effects of SGLT2i and GLP1ra on primary prevention of CKD in patients with T2D. The evidence looks strong because of a large number of participants. However, I have some concerns and suggestions related to the study.

  1. To design of the study (Fig. 1):

1.1. The authors did not note the period between two consequent records of abnormal eGFR and UACR. Please ensure that it was at least 3 months according to the CKD definition by KDIGO.

Response: Thanks to the reviewer for raising this question. Our source of information was an electronic healthcare database comprised of electronic health records from primary care, so it is important to note that the data that contains is the recorded by the physicians in their daily routine for clinical purposes rather than research purposes. Taking that into account, we must fit our ideal methods and research objectives to the data available. In this sense, in previous analyses of our cohort, we found that levels of serum creatinine or eGFR were recorded in the database for each subject every year, on average, so we could be confident that such time interval between measurements occurred also in our cases.

We propose to add the following text at the end of “Discussion”, among the limitations, at page 10, lines 271-275:

Although, we were unable to confirm that the two consecutive abnormal levels were separated by at least 90 days, in previous analyses of our cohort we found that levels of serum creatinine or eGFR were recorded in the database for each subject every year, on average, so we could be confident that such time interval between measurements occurred also in our cases.”

1.2. The manuscript does not contain such important outcomes as death etc.

Response: Thanks for the comment. Death was not an outcome of interest in the present study as we aimed to study the drivers of CKD in the cohort. To study death as an outcome in the case-control analysis, we would have to extend the follow-up to death among cases and to conduct another case-control study (using death as outcome and not CKD), and that was not within the objectives of the present study.

No changes are proposed

  1. To statistical procedures: The authors analyzed a lot of data. May be it could be reasonable to count p-value for AOR and analyze the significance assuming multiple testing problem.

Response: Thanks to the reviewer for the comment. We prefer to report confidence intervals (CIs) instead of p-values as CIs provide an estimation of the population effect size and give the reader a more comprehensive view of the data (see Tijssen JGP, More confidence intervals and fewer p values JACC 2021; 1562-63). Additionally, experts in epidemiological methods discourage to apply adjustments for multiple comparisons, among other reasons because, as they are intended to reduce the type I error, such methods increase the type II error and may preclude the possibility of observing true associations (see Rothman K, No adjustments are needed for multiple comparisons, Epidemiology, 1990, 1:43-46; http://www.jstor.org/stable/20065622?origin=JSTOR-pdf)

No changes are proposed

  1. To interpretation of results: The authors proved the beneficial effects of SGLT2i and GLP1ra. However, the manuscript contains some questionable results, such as harmful effects of all other GLDs on CKD incidence. Why did the authors include these GLDs to analysis while they aimed to study effects of SGLT2i and GLP1ra? What could be the possible cause of this observation? Which flaws in the design of the study or statistical analysis could bias the results?

Response: Thanks to the reviewer to raise such important question. In this study, all patients conforming the cohort are diabetic patients on treatment with at least one GLD. Thus, there is no group without treatment and each pharmacological subgroup analyzed is compared to all others. So, the interpretation of our results is that long-term use of SGLT-2i and GLP-1RA, as compared to the use of all other GLDs, show an association with a reduced risk of an incident CKD. We do not feel, and do not state in the paper, that the other GLDs are associated with an increased risk, rather, we hypothesize that they are not effective controlling the onset and progression of renal damage associated to diabetes.

We included an additional text in the “Conclusions”, page 13, lines 407-409, to clarify this question as follows: “… a finding which is in line with post-hoc analyses of clinical trials of these agents. In addition, the majority of the remaining GLDs analyzed did not show evidence supporting their use for the primary prevention or to control the progression of CKD, especially at long-term

  1. To presentation (Tab. 2): It is not clear what means the lines 1 (1-1) and 1 (1-2). Please, explain.

Response: Thanks to the reviewer for the comment. In the Table 2, that numbers associated to prior records of eGFR or albuminuria correspond to the median number of records and percentiles 25 and 75.

Reviewer 2 Report

Comments and Suggestions for Authors

Overview The authors' team used a retrospective cohort study of nested case-control analysis to assess the incidence rate of CKD, risk factors, and the impact of hypoglycemic drugs on primary prevention of CKD. The research is innovative, and the methods are scientifically rigorous, but there are still a small number of issues that need to be addressed.   Details 1. Clinical research should have ethical and informed consent certificates. If necessary, clinical registration is also required. 2. The inclusion criteria for Figure 1 are very detailed, but the exclusion criteria are too simple. All possible scenarios should be listed in detail. In addition, are there detailed inclusion and exclusion criteria also for the control group? Or should we use the same standards overall? This figure cannot be distinguished. 3. Figure 2 does not seem to be the main content of the article and should not be listed as a separate section, or the author should provide more trend information as a common finding. In short, writing about the incidence of gender and age separately does not seem to be the main issue.   4. The two supplementary tables seem to be the main content, and we believe they should be added to the main part of the article.

Comments on the Quality of English Language

Minor editing of English language required.

Author Response

Reviewer 2

Overview The authors' team used a retrospective cohort study of nested case-control analysis to assess the incidence rate of CKD, risk factors, and the impact of hypoglycemic drugs on primary prevention of CKD. The research is innovative, and the methods are scientifically rigorous, but there are still a small number of issues that need to be addressed.   Details 

  1. Clinical research should have ethical and informed consent certificates. If necessary, clinical registration is also required. 

Response: Thank to the reviewer for the comment. We used an electronic healthcare database where personal data was pseudonymized and subjects are unidentifiable by the researchers. In addition, we had a cohort of almost 6 million subjects so the efforts to obtain an informed consent (if that could be possible) would exceed the benefits of the research. In such cases, the Spanish and European regulation on personal data permits a waiver to the informed consent. In addition, the ethics committee of the Hospital “12 de Octubre” (Madrid, Spain) reviewed and approved the study protocol including such waiver for the informed consent.

We state that in page 14, sections “Institutional Review Board Statement” and “Informed Consent Statement”, lines 454-459, so no additional changes are proposed.

  1. The inclusion criteria for Figure 1 are very detailed, but the exclusion criteria are too simple. All possible scenarios should be listed in detail. In addition, are there detailed inclusion and exclusion criteria also for the control group? Or should we use the same standards overall? This figure cannot be distinguished.

Response: Thanks to the reviewer for the comment. In Figure 1 we only showed those inclusion and exclusion criteria for the case-control analysis that were additional to those applied for the general cohort. In the case-control analysis, both cases and controls must fulfill the same inclusion and exclusion criteria to ensure they were extracted from the same population. That is also relevant to interpret the measure of association as an estimation of the measure of association that would be obtained in a cohort analysis.

Changes proposed: We included in Figure 1 all inclusion and exclusion criteria, not just those applied to the case-control analysis. In consequence, a new Figure 1 was submitted.

  1. Figure 2 does not seem to be the main content of the article and should not be listed as a separate section, or the author should provide more trend information as a common finding. In short, writing about the incidence of gender and age separately does not seem to be the main issue.   

Response: Thanks to the reviewer for the advice. One of the objectives of the study was to study the drivers of CKD in a general cohort. It is widely known that gender and age are important risk factors of CKD, so in this line, we showed incidence rates of CKD in a cohort of subjects with type-2 diabetes using data from the real-world in primary care, by gender and age. Although we agree that these data are purely descriptive, we feel that may be of interest for the readers. However, we would accept to pass it to the supplement if required.   

We do not propose any change, but we are open to them on editor’s demand.

  1. The two supplementary tables seem to be the main content, and we believe they should be added to the main part of the article

Response: Thanks to the reviewer for the comment. The two supplementary tables are an extension of Figures 4 and 5. Due to figure constraints we preferred to show the relevant data in the main figures and expand the information in table format as a supplement. In consequence, we do not deem necessary to pass the supplementary tables to the main text as they would be redundant.

We do not propose any changes.

Reviewer 3 Report

Comments and Suggestions for Authors

Line 86 - electronic, not electronical.

Line 88 - can you say "causal" in an observational study?

Line 101 - remove "the".

Line 123 - "conform" - better "to make up".

Line 162 - on line 123 you write everyone was treated with a GLD. Here you chose persons not on GLD. Please explain this contradiction. Please explain lines 166-167 - I do not understand.

Line 189 - especially.

Line 205 - made up.

The authors mention that testing for ALB is not high. What percent of the total cohort had ALB testing? that is - how many were excluded because of missing data?

Comments on the Quality of English Language

English is fine  - just a few mistakes

Author Response

Reviewer 3

  1. Line 86 - electronic, not electronical.

Line 101 - remove "the".

Line 123 - "conform" - better "to make up".

Line 189 - especially.

Line 205 - made up.

Response: Thanks to the reviewer for the corrections. We reviewed the main text, so all errors have been amended.

  1. Line 88 - can you say "causal" in an observational study?

Response: Thanks to the reviewer for the comment.

The reviewer is right that only randomized clinical trials can state “causal effects”, and this even with limitations (intention-to-treat, no withdrawals, or perfect masking, among others). But it is undeniable that observational research may explore causal effects and support or put in question causal effects, especially when clinical trials are unpractical or unethical. Let us refer to the important observational studies that discovered that smoking was a cause for lung cancer, among many other important public health issues supported by observational research.

Having said that, we propose to remove the word “causal” in the “Introduction”, page 2, line 88, to avoid any potential confusion.

  1. Line 162 - on line 123 you write everyone was treated with a GLD. Here you chose persons not on GLD. Please explain this contradiction. Please explain lines 166-167 - I do not understand.

Response: Thanks to the reviewer to raise that question. All subjects in the cohort were type-2 diabetics under treatment with at least one glucose-lowering drug (GLD). Then, what we did is to compare the use of an individual GLD with all other GLDs. The non-user category which appears in “Nested case-control study”, lines 359-365, refers to no prescriptions of that individual GLD, but necessarily on treatment with any other GLD.  

We stated that clarification in “Patients and Methods”, page 12, lines 363-365: “The reference category for all comparisons was the non-use. As all subjects were users of GLDs, the non-use category can be interpreted as “current use of any other GLDs”.

Round 2

Reviewer 1 Report

Comments and Suggestions for Authors

The authors had made the changes in the manuscript as well as replied to the Reviewer comment. However, I do not consider that answer to the comments 4 and 5 were sufficient.

To the Comment 4: Fig. 4 and  5 display the core results. According to the Fig. 4 and 5, it seems that all GLDs excluding SGLT2i and GLP1ra >3 yr are harmful for kidney. The detailed explanation for these graphs could be considered to add to the figure footnote or the text. What did the authors compared? Was it the patients who received these agents and others T2D patients? Assuming the inclusion criteria (at least one GLD, but not only insulin) and relatively few number of participants received SGLT2i and GLP1ra (i.g. 854 and 1188, respectively, >3 yr of ~500K participants in Fig. 4 and 344 and 461, respectively, >3 yr of ~250K participants in Fig. 5), it is not clear why all other GLDs were significantly worse for renal outcomes.

The Comment 5: Formally, the notes 1 (1-1) or 1 (1-2) [Me, IQR] are correct. However, it is data set with narrow range and it could be more understandable to present these results as Min-Max or numbers and percentages for 0 prior records, 1 prior record, 2 prior records etc. on the authors choise.

Author Response

The authors had made the changes in the manuscript as well as replied to the Reviewer comment. However, I do not consider that answer to the comments 4 and 5 were sufficient.

To the Comment 4: Fig. 4 and  5 display the core results. According to the Fig. 4 and 5, it seems that all GLDs excluding SGLT2i and GLP1ra >3 yr are harmful for kidney. The detailed explanation for these graphs could be considered to add to the figure footnote or the text. What did the authors compared? Was it the patients who received these agents and others T2D patients? Assuming the inclusion criteria (at least one GLD, but not only insulin) and relatively few number of participants received SGLT2i and GLP1ra (i.g. 854 and 1188, respectively, >3 yr of ~500K participants in Fig. 4 and 344 and 461, respectively, >3 yr of ~250K participants in Fig. 5), it is not clear why all other GLDs were significantly worse for renal outcomes.

Response: Thanks to the reviewer for the comment. Although we partially share the point of view of the reviewer, we cannot make interpretations that would fall beyond the scope of our methods and results. We strongly believe that the long-term use of SGLT-2i and GLP-1 RAs are protective factors of incident CKD, as observed in our data and also supported in clinical trials. Of note, such protective effect was observed within T2DM patients with the longest durations of the disease, on average (supplementary tables 1 and 2), and probably these patients received SLGT-2i and GLP-1 RAs as second or third line of treatment, thus with higher risk of develop CKD. So, if SGLT-2i and GLP-1RAs didn’t have any effect we should have observed an increased risk of CKD. That is precisely what we hypothesize that occurs among the users of the rest of GLDs, that is the measure of association observed may reflect the effect of time of duration of T2DM, maybe not well controlled, also supported by the fact that they have not demonstrated to improve the kidney function in clinical trials.

We propose to include in the Discussion, page 10, lines 246-250, the following statement: “Of note, users of SGLT-2i and GLP-1 RAs had the longest time of duration of type 2 diabetes, on average, so they were likely at a higher risk of CKD. If SGLT-2i and GLP-1RAs did not have any effect we should have observed an increased risk of CKD as we did among the users of some of the remaining GLDs, especially at long-term, the latter maybe reflecting the lack of such improvement of kidney outcomes”

We also propose to include a footnote in Figures 4 and 5, pages 8 and 9, to clarify the category of reference for each comparison.

The Comment 5: Formally, the notes 1 (1-1) or 1 (1-2) [Me, IQR] are correct. However, it is data set with narrow range and it could be more understandable to present these results as Min-Max or numbers and percentages for 0 prior records, 1 prior record, 2 prior records etc. on the authors choise.

 Response: Thanks for the comment. We agree with the reviewer, so to increase the interpretability of results we switched ranges for the number and percentage of subjects with more than 1 prior record.

We included those changes in Table 2, page 6

Round 3

Reviewer 1 Report

Comments and Suggestions for Authors

Have reviewed the revised version of manuscript, I cannot add anything because the authors replied the comments and stated that the calculations and results were correct.

However, due to plenty of assumed cofounding factors, including medications and some other states I would ask the authors to add the Results or Supplementary materials by the information about completeness of filling the data and possible limitations when used this database.

I also noted that the authors adjusted risk of CKD or declined eGFR on "prior isolated pathological records of eGFR, prior isolated pathological records of proteinuria or albuminuria" (Supplementary Methods section). Maybe it would be better to exclude this cofounder.

Author Response

Ref.:

Manuscript title: “GLUCOSE-LOWERING DRUGS AND PRIMARY PREVENTION OF CHRONIC KIDNEY DISEASE IN TYPE 2 DIABETES PATIENTS: A REAL-WORLD PRIMARY CARE STUDY

Dear Editors of Pharmaceuticals:

Thank you for giving us the opportunity to resubmit a revised version of our paper. Below you will find the point-by-point answers to the additional comments raised by the reviewer 1. Also, all changes performed in the original text have been highlighted conveniently.

Finally, you will find a file with the responses to reviewers and the amended version of the paper with tracked changes.

Reviewer 1 (Round 3)

Have reviewed the revised version of manuscript, I cannot add anything because the authors replied the comments and stated that the calculations and results were correct.

However, due to plenty of assumed cofounding factors, including medications and some other states I would ask the authors to add the Results or Supplementary materials by the information about completeness of filling the data and possible limitations when used this database.

I also noted that the authors adjusted risk of CKD or declined eGFR on "prior isolated pathological records of eGFR, prior isolated pathological records of proteinuria or albuminuria" (Supplementary Methods section). Maybe it would be better to exclude this cofounder.

Response: Thanks to the reviewer for the comments.

BIFAP is a healthcare electronic database comprised of electronic medical records from, mainly, primary care. The information contained in those records is the recorded by physician in their daily routine and not for research purposes. Lack of randomization in observational research often leaves room for bias such as confounding, maybe more marked in a case-control study where cases must be widely different from controls. However, we thoroughly investigated about potential sources of bias (including confounding, selection and missing data) before specification of the model.  All potential confounders were selected based on expert knowledge and not based on data driven methods. With that being said, we strongly believe that prior pathological records of eGFR or proteinuria/albuminuria act as a confounder as it predicts the exposure, since the decision to start or switch the treatment with GLDs must be based on a potential kidney decline, and the outcome as well, since prior pathological records or acute kidney diseases increase the risk for further CKD. Within this framework, excluding such covariates from the model could open a backdoor path, introducing a biased association between the exposure and the outcome even if it does not exist.

We described our source of information in page 11, “Patients and Methods; Source of Information”, and we widely stated the abovementioned limitations of the study in the “Discussion”, page 9, so no additional changes are proposed.

We are unaware about the rationale of the reviewer to not include such covariates in the model, but in case overfitting was the main concern, we performed a sensitivity analysis excluding such covariates from the model, and results did not vary. We only show fully-adjusted odds ratios for our main results, as follows:

Incident CKD overall:

SGLT-2i:

<3 years: 1.07 (1.02-1.12)

≥3 years: 0.88 (0.73-1.06)

GLP-1RAs:

<3 years: 0.95 (0.89-1.02)

≥3 years: 0.84 (0.72-0.98)

Incident CKI (G3-G5):

SGLT-2i:

<3 years: 1.05 (0.97-1.13)

≥3 years: 0.65 (0.46-0.91)

GLP-1RAs:

<3 years: 0.92 (0.82-1.04)

≥3 years: 0.73 (0.57-0.95)

In addition, computing Akaike’s information criterion (AIC) and Bayesian information criterion (BIC) as measures of the balance between fitting and complexity of maximum likelihood models, yielded lower scores for the model that includes such covariates compared to the model without them, thus tipping the balance towards the most complex model:

Incident CKD overall model

With: AIC (289089.7); BIC (290563.5)

Without: AIC (291114.6); BIC (292566)

Incident CKI (G3-G5) model

With: AIC (129580.7); BIC (130963)

Without: AIC (131987.9); BIC (133349.2)

No additional changes are proposed about results of the study.
